# Factors Associated with Psychological Distress in French Medical Students during the COVID-19 Health Crisis: A Cross-Sectional Study

**DOI:** 10.3390/ijerph182412951

**Published:** 2021-12-08

**Authors:** Carole Pelissier, Manon Viale, Philippe Berthelot, Brigitte Poizat, Catherine Massoubre, Theophile Tiffet, Luc Fontana

**Affiliations:** 1Univ Lyon, Univ Lyon 1, Univ St Etienne, University Gustave Eiffel-IFSTARR, UMRESTTE, UMR_T9405, 42005 Saint-Etienne, France; luc.fontana@chu-st-etienne.fr; 2Occupational Health Service, University Hospital Center of Saint-Etienne, 42005 Saint-Etienne, France; manon.viale@gmail.com; 3Infection Control Unit, Infectious Diseases Department, University Hospital of Saint-Etienne, 42000 Saint-Etienne, France; philippe.berthelot@univ-st-etienne.fr; 4Preventive Medicine Department, Jean Monnet University, 42000 Saint-Etienne, France; brigitte.poizat@univ-st-etienne.fr; 5Department of Psychiatry, University Hospital Center of Saint-Etienne, 42005 Saint-Etienne, France; catherine.massoubre@chu-st-etienne.fr; 6Public Health Service, University Hospital Center of Saint-Etienne, 42005 Saint-Etienne, France; Theophile.Tiffet@chu-st-etienne.fr

**Keywords:** psychological distress, medical students, COVID-19 health crisis, traumatic event, distance learning

## Abstract

Background: The purpose of this study was to assess the prevalence of psychological distress in medical students during the COVID-19 health crisis and to identify factors associated with psychological distress. Methods: A cross-sectional observational study was presented to 1814 medical students (from first to sixth year) in a French university hospital center. Sociodemographic, occupational and medical information (psychological distress measured on the French GHQ12 scale) were collected via an online anonymous self-administered questionnaire. Variables associated with psychological distress were investigated using univariate analysis and multivariate analysis (modified Poisson regression). Results: In total, 832 medical students responded (46%) and 699 completed the questionnaire in full (39%); 625 (75%) showed signs of psychological distress and 109 (15%) reported suicidal ideation. Female gender, psychological trauma during the COVID-19 health crisis, change in alcohol consumption, and difficulties with online learning emerged as risk factors for psychological distress, whereas a paid activity, a feeling of mutual aid and cooperation within the studies framework, and recognition of work appeared to be protective factors. Conclusions: Mental health care or suicide prevention should be provided to students at risk in the aftermath of the pandemic. Knowing the educational and medical factors associated with psychological distress enables areas for prevention to be identified.

## 1. Introduction

Since the beginning of 2020, SARS-CoV-2 has spread to several continents and is responsible for a large number of deaths [1,2]. To reduce the risk of person-to-person viral transmission during the COVID-19 pandemic, the French government introduced various measures, including social distancing, self-quarantine, and temporarily cancelling work and school, to control the disease. Students are bearing the brunt of the economic, social and psychological consequences of the COVID-19 pandemic. There is growing concern worldwide regarding the psychological health of students and particularly of medical students. Medical schools around the world have long been considered stressful environments for students entering higher education [3]. Students enter medical school immediately after high school, often at 18 years of age, and they go through 6 years of medical education before graduation. In France, medical students take a competitive examination at the end of their first year and a national competitive examination at the end of their sixth year. During the first 3 years, students have preclinical training, then from 4th to 6th year they have clinical training. Previous research identified long hours of study, academic workload, competition with peers, conflicts in work-life balance, the emotional burden of exposure to human suffering, and considerable financial pressure as the principal stressors affecting psychological health [4,5,6,7,8]. Psychological distress broadly refers to anxiety, stress, depression, and mental health-related problems. Previous studies showed that the prevalence of psychological distress in medical students during medical training in various countries and institutions ranges from 21% to 56% [9]. Even before the COVID-19 pandemic, medical students showed higher rates of mental health issues than the general population, including generalized anxiety disorder (GAD), depression, and burnout [10,11]. Yusoff et al. reported that healthy students develop depression and stress after commencing their medical education [12]. It was also reported that physicians tend to have higher suicide rates compared to the general population [13]. 

Previous pandemics, such as the 2003 SARS outbreak in China, increased stress levels in healthcare students, highlighting the need for additional support for this population during public health crises [14]. The spread of the COVID-19 virus has had far-reaching consequences, and the closure of universities has led to the development of online learning, leading to student isolation. Students experiencing higher psychological distress are at a higher risk of academic failure and dropout [15,16]. In addition, medical students during internships in COVID-19 care units were involved in the management of patients infected with COVID-19 which may have exposed them to a high emotional burden.

We assume that the organizational changes in theoretical (distance learning courses) and practical (internships in COVID-19 care services) teaching caused by the health crisis are associated with psychological distress in medical students.

The objectives of the present study were to evaluate the prevalence of psychological distress in medical students during the COVID-19 health crisis and to identify personal, medical and occupational factors associated with psychological distress.

In this context, an evaluation of the prevalence of psychological distress and the associated factors in medical students in France was conducted.

## 2. Materials and Methods 

The study design consisted of a cross-sectional questionnaire survey.

### 2.1. Target Population

Data were collected from 10 March to 25 March 2021. The target population was 1st- to 6th-year students registered at the Faculty of Medicine of Saint Etienne located in the Loire department which recorded the highest incidence rate of COVID-19 in the autumn of 2020 (700 cases per 100,000 inhabitants) [17]. This epidemic situation has led to an increase in the demand for care in the hospitals of the Loire department. Medical students in hospital internships were involved in the health crisis by participating in the care of patients infected with COVID-19. Students from 1st to 3rd year may have been victims of social isolation due to the distance learning course and the implementation of a lockdown period from 30 October 2020 to 15 December 2020. 

### 2.2. Study Sample

We targeted medical students at different levels of training, at preclinical (first and second year) or clinical level (third to sixth year). All students aged at least 18 years old and registered in medicine at Saint Etienne medical school from 1st to 6th year were invited to respond voluntarily to a self-administered online survey. They received an invitation to participate in this study via their e-mail address. In total, 1814 eligible students were contacted by university email. The participants answered the online questionnaire via the LimeSurvey application (Hamburg, Germany). They received clear and comprehensible information on study objectives and procedure, and were free to decline participation. Review board approval (IRBN272021/CHUSTE) was obtained before starting the study. The average age of the eligible subjects in the study was 21 years old [Et = 1.17]; 66% were women, 65% were 1st-year students, 19% were 2nd- and 3rd-year students, and 26% were 4th-, 5th- and 6th-year medical students.

### 2.3. Measurements

We developed a self-reported questionnaire to collect data on demographic, occupational and medical characteristics. Self-administration time was measured to approximately 10 min.

The main endpoint (psychological distress) was assessed by the validated French version of the 12-item General Health Questionnaire (GHQ) [18], a self-report instrument measuring psychological morbidity, intended to detect psychiatric disorders in community and non-psychiatric settings [19]. The Cronbach coefficient of the GHQ-12 was evaluated by Goldberg at between 0.82 and 0.86 in general health care [20]. The Cronbach coefficient of the GHQ-12 was evaluated at 0.85 in a population of Malian students [21]. Answers were given on a 4-point scale; for instance, the item “In the last weeks, did you feel under strain?” allows for the following answers: “No”, “Not more than usual”, “More than usual”, and “Much more than usual”. When scored with the binary method (0–0–1–1), the GHQ-12 can be used as a screening tool to detect minor non-psychotic psychiatric disorders, yielding final scores that range from 0 to 12. Operationally, patients scoring ≥4 are considered “GHQ-positive” [22]. 

The anonymous self-administered questionnaire covered 3 areas.

Personal: gender, age, number of people at home, and financial difficulties.

Educational: seniority in medical studies, weekly study time, daily screen time, face-to-face courses in the last three months, hospital internship in the last three months, internship in COVID-19 care services, and difficulties in following distance learning courses. Perceived stress related to personal and educational life was assessed on a visual analogue scale (VAS). A cut-off at 7 points defined clinical signs suggestive of stress. 

Medical: perceived health status, experience of trauma during the COVID-19 crisis, sense of mutual aid and cooperation in studies, history of anxiety disorder, history of depression, history of suicide attempts, psychotropic treatment, psychiatric care, presence of suicidal ideation, date of last consultation with general practitioner or occupational/prevention physician, sleep duration, alcohol consumption and smoking, and cannabis use.

### 2.4. Analysis

Age and stress levels, both quantitative variables, were transformed into categorical qualitative variables before statistical analysis.

A descriptive analysis was made of the sample’s sociodemographic, educational and medical characteristics. We chose not to investigate the association between psychological distress and suicidal ideation since suicidal ideation appears to be a complication of psychological distress.

A univariate analysis was performed to assess the association between psychological distress and sociodemographic, educational and medical factors. Chi^2^ and Fisher tests were applied as appropriate. The significance threshold was set at 5%. Variables significantly associated with psychological distress were introduced in a modified Poisson regression using robust variance estimations [23]. Variables with a *p*-value ≤ 0.2 were included in the multivariate model, and variables with a *p*-value < 0.05 were kept in the model. Analyses used R software (The R Foundation for Statistical Computing, Vienna, Austria) used in France.

## 3. Results

### 3.1. Sociodemographic, Educational and Medical Characteristics

As shown in Table 1 and Table 2, out of the 1814 eligible students, 832 (73% female, 27% male) responded, giving a response rate of 46% (Figure 1). More than a third of respondents were 19 or 20 years of age. Three quarters were single. Nearly 10% reported financial difficulties. Half expressed high levels of stress related to their personal life. Nearly one third reported psychological trauma related to the COVID-19 health crisis. A minority reported increased smoking, alcohol use, or cannabis use. More than a quarter reported sleeping less than 6 hours per night. Nearly half reported never having seen a general practitioner. More than half said they spent more than 40 hours a week studying. More than three quarters reported a very high level of study-related stress. The majority reported difficulties related to personal and occupational time management. However, half reported helping each other and nearly two thirds reported recognition of their work. Nearly 15% reported suicidal ideation and 4% reported a suicide attempt.

### 3.2. Prevalence of Psychological Distress 

A total of 625 respondents (75%) presented psychological distress. 

### 3.3. Relations between Psychological Distress and Educational and Medical Factors on Univariate Analysis 

A search for multicollinearity was conducted for the following variables: age, gender, financial difficulties, trauma experienced during the COVID-19 crisis, history of anxiety disorders, history of depression, changes in alcohol consumption, changes in smoking, paid work outside the framework of studies, sense of mutual support and cooperation, impression of recognized work, hospital internship in the last 3 months, internship in COVID-19 care units, difficulties in following distance learning courses, and date of last consultation with a general practitioner. Following this analysis, it was decided to remove the variable for an internship during the last three months due to a significant multicollinearity with other variables.

As shown in Table 3, the univariate analysis produced associations between psychological distress and the following:Female gender, PR = 1.12 (1.02–1.23)Age, PR = 0.98 (0.96–0.99);1st year of medical school (major), PR = 1.11 (1.01–1.21);Financial difficulties, PR = 1.11 (1.01–1.22);Psychological trauma during the health crisis COVID-19, PR = 1.21 (1.11–1.29);History of anxiety disorder, PR = 1.12 (1.03–1.22);History of depression, PR = 1.13 (1.03–1.24);Change in smoking, PR = 1.08 (0.99–1.19);Change in alcohol consumption, PR = 1.08 (1.01–1.16);Sense of mutual support and cooperation, PR = 0.80 (0.75–0.86);Impression of recognized work, PR = 0.75 (0.69–0.82);Hospital internship within the last three months, PR = 0.91 (0.84–0.98);Hospital internship on a COVID-19 ward within the last three months, PR = 0.90 (0.81–0.99);Experiencing difficulties with online learning, PR = 1.64 (1.39–1.94).

In contrast, psychological distress was not significantly associated with increased cannabis use or weekly study workload. 

### 3.4. Relations between Psychological Distress and Educational and Medical Factors in Multivariate Analysis 

As shown in Table 3 and Figure 2, in the multivariate analysis, psychological distress remained associated with female gender, a history of anxiety disorders, psychological trauma during the health crisis, change in alcohol consumption, and difficulties in online learning. The feeling of mutual aid and cooperation within the studies framework and work recognition appeared to be protective factors.

## 4. Discussion

Medical students have been shown to be at higher risk of mental health disorders during training [24]. Although medical students have better access to mental health care, they are less likely to seek help than the general population, mainly due to the stigma attached to mental health disorders [25]. Our study showed a 75% prevalence of psychological distress in medical students in years 1–6. This rate was higher than for Essangri et al. in a cross-sectional online survey conducted from 8 April to 18 April 2020 which showed a 69% prevalence of psychological distress in medical students in Morocco. Meta-analyses indicated that 9–54% of students worldwide experience psychological distress. These differences can be explained by the context of the health crisis: lockdown, distance learning, fear of contracting COVID-19, and social insecurity. Students may be consumed by major uncertainties regarding their future and educational perspectives. Distance education and examinations may increase their level of uncertainty and stress, either because these involve new and unfamiliar teaching and assessment modalities or because distance supervision, communication, and monitoring by teachers has not been sufficiently clear, structured, or reassuring [26].

Our results showed greater risk of psychological distress in women. These findings are consistent with the literature. Previous mental health research highlighted female gender as a vulnerability factor for poorer mental health and well-being [27].

Nearly one third of medical students reported a traumatic event during the COVID-19 health crisis. This rate was lower than that reported by Waseem et al., who found that more than half of Pakistani medical students showed moderate to severe post-traumatic stress (54.10%) [28]. These differences could be explained by the alarming increase in the number of COVID-19 cases in Pakistan during the period in which the study was conducted. Our study showed an association between traumatic events and psychological distress, which is consistent with the findings of Lasalvia et al. that half of the 2195 healthcare workers who reported a COVID-19-related traumatic experience also reported symptoms of clinically significant anxiety [29].

Similar to Clay and Parker’s, our findings showed that a larger percentage of students decreased than increased their alcohol consumption [30]. Young people mainly use alcohol in social contexts, and drink alcohol less often and in smaller quantities but with an anxiolytic effect [26]. However, our study highlighted the significant association between psychological distress and change in alcohol consumption. This significant association was maintained after adjustment for other co-variates. These results are consistent with those of Lechner et al., who showed that more severe psychological distress in students was associated with higher alcohol consumption overall. These results underline the value of the early detection of increased alcohol consumption in the prevention of psychological distress in students. 

The difficulties associated with distance learning emerged as a risk factor for students’ psychological distress. The closure of universities and public libraries and the limited access to alternative study spaces forced many students into an unaccustomed learning environment [31]. The rapid change in the system and environment could cause significant stress to medical students [32]. Due to the pandemic, which forced educational institutions to eliminate in-person teaching sessions, medical students needed to adapt to new educational environments, such as distance e-learning [33]. The rapid change in the system and environment could cause significant stress to medical students [34]. Turning to distance learning on a global scale leads to a risk of exacerbating educational inequalities, jeopardizing students’ mental health [35]. Accumulating evidence suggests that mindful coping effectively reduces stress and anxiety in college students. Improvements in self-esteem and self-efficacy would strengthen resilience and motivation towards learning and career development [36].

Students also reported difficulties with distance learning due to time management, personal life/working life balance, workload, and a lack of communication with teaching staff. Our findings highlighted the protective effect of social support against psychological distress. These results corroborate those of previous studies [37,38]. Psychological support should be tailored to each student’s needs and incorporated into the online remote curriculum [39].

The study also highlighted how mutual help in studies has a protective effect against psychological distress. Low perceived social support was significantly associated with an increased risk of psychological distress [40]. According to Cao et al., people with low perceived social support were at high risk of psychological pressure, while high perceived social support had a positive effect on anxiety and stress during the COVID-19 epidemic [41]. Moreover, poor esteem due to superiors emerged as a risk factor for psychological distress [42,43]. In our study, recognition of the work performed appeared to be a protective factor against psychological distress. This result emphasizes the benefits of supportive communication in preventing psychological distress in students.

Moreover, our study highlighted how maintaining occupational activity has a protective effect against psychological distress. These results corroborate those of previous studies. According to Essadek et al., being in a precarious financial situation significantly increased levels of depression, anxiety, and distress [44]. As shown in previous studies, there may be an increasing prevalence of food insecurity during the pandemic, negatively affecting students’ mental well-being [34,45,46]. To prevent financial insecurity for students, the French government implemented a policy of financial aid (meals at €1 and exceptional aid in case of job loss).

### Strengths and Limitations

The strengths and limitations of the current study are determined by several issues. We collected data from one medical school only; this may be a somewhat unbalanced sample that does not fully represent the diversity of medical students currently in training in France. In addition, the sample in our study has a slightly different distribution from the source population with regard to gender (73% women in the sample, 66% in the population) and years of study (60% first-year students in the sample, 56% in the population). These differences in distribution may contribute to an overestimation of psychological distress. Indeed, according to our study, female gender appears to be a factor positively associated with psychological distress, while age appears to be negatively associated with psychological distress. These results are consistent with the study by Maser et al. of Canadian medical students [47]. Furthermore, 45% of the students enrolled at Saint-Etienne medical school agreed to participate in this study. Of these, 92% completed the GHQ-12 and 84% completed the questionnaire in full. The refusal and dropout rates of this study should be considered before generalizing the findings of this study.

Moreover, we adopted a convenient online survey in only one university in France, which may contribute to some bias in the study results. The e-questionnaire assessed the prevalence of psychological distress in university students adhering to WHO-recommended “social distancing” during the COVID-19 pandemic. The e-survey data were collected by globally validated standardized tools for quantitative analysis. In this cross-sectional study, the identified factors were regarded as associated factors, which could either be the causes or the results of psychological distress. Furthermore, due to the ethical requirements of anonymity and confidentiality, the contact details of the respondents were not collected. However, the use of a validated screening e-questionnaire was considered to be a cost-effective approach to explore the situation in general, and was therefore used in this study. Since the research methodology could not reach people with psychological distress under treatment, the results may not fully reflect the severity of psychological distress symptoms in students. A follow-up study could follow up the same participants to determine the persistence or transience of the perceived psychological distress. Mental health problems in medical students need to be further assessed longitudinally.

## 5. Conclusions

The high prevalence of psychological distress among medical students observed in our study shows the importance of promoting early detection by preventive and occupational medicine services and facilitating psychological management by psychologists. It is suggested that the government and universities should collaborate to resolve this problem and provide high-quality and timely crisis-oriented psychological services to medical students [48]. This care should be based on the implementation by the French government, since 10 March 2021, of a national platform of psychological support for students; sessions with a psychologist, of up to three meetings of 45 min each, are completely free. The identification of risk factors and protective factors for psychological distress can determine adapted means of preventing psychological distress in medical students. The deployment of distance learning should be based on pedagogical support, including frequent exchanges with teachers and other students. Longitudinal follow-up studies are required to track the progression of psychological distress in medical students and measure the long-term impact of the pandemic.

## Figures and Tables

**Figure 1 ijerph-18-12951-f001:**
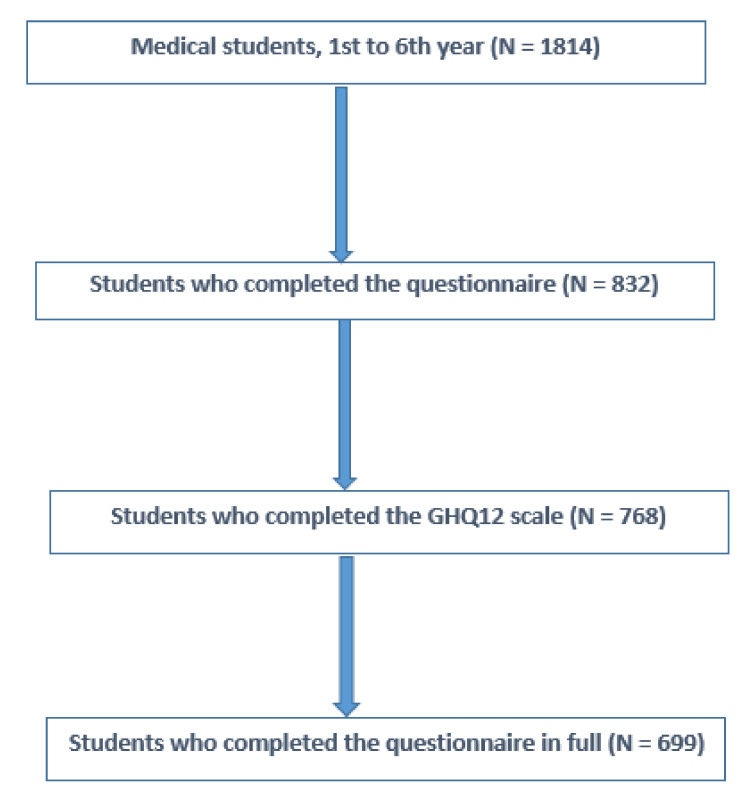
Flowchart of the studied population.

**Figure 2 ijerph-18-12951-f002:**
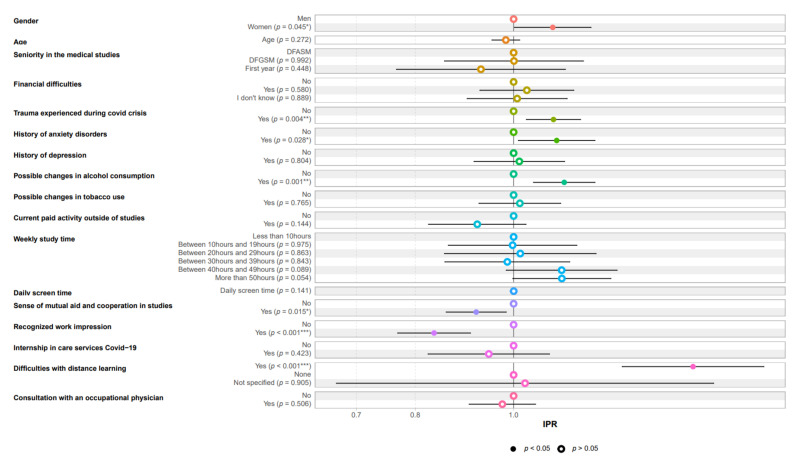
Factors associated with psychological distress in multivariate analysis.

**Table 1 ijerph-18-12951-t001:** Medical characteristics.

				Psychological Distress			
			Total	Yes (N = 625, 75.1%)		No (N = 207, 24.9%)	
		N	%	N	%	N	%
Gender	Women	609	73.2	470	77.2	139	22.8
	Men	223	26.8	155	69.5	68	30.5
Age	18 years	248	29.8	197	79.4	51	20.6
	19–20 years	339	40.7	248	73.2	91	26.8
	21 years and over	244	29.4	180	73.8	64	26.2
Family situation	Single	623	75.0	466	25.2	157	74.8
	In a couple	203	24.4	155	76.3	48	23.7
	Widowed, Separated, Divorced	5	0.6	4	80.0	1	20.0
Living alone	Yes	32	3.8	22	68.8	10	31.2
	No	799	96.1	196	24.5	603	85.5
Financial difficulties	Low	684	84.3	519	75.9	165	24.1
	High	78	9.6	66	84.6	12	15.4
	Don’t know	49	6.0	40	81.6	9	18.4
Perceived health status	Poor to mediocre	99	13.3	97	98.0	2	2.0
	Moderate	368	49.6	322	87.5	46	12.5
	High, very high	275	37.1	183	33.4	92	66.5
Perceived stress level	Low to moderate	368	49.6	251	68.2	117	31.8
	High	374	50.4	351	93.9	23	6.1
Trauma experienced during the COVID-19 crisis	No	510	68.7	388	76.1	122	29.9
	Yes	232	31.3	214	92.2	18	7.8
History of anxiety disorders	No	650	87.6	519	79.8	131	20.1
	Yes	92	12.4	83	90.2	9	9.8
History of depression	No	688	92.7	553	80.4	135	19.6
	Yes	54	7.3	49	90.7	5	903
Presence of suicidal ideation	No	631	85.2	494	78.3	137	21.7
	Yes	109	14.7	106	97.3	3	2.7
History of suicide attempts	No	684	95.9	549	80.3	135	19.7
	Yes	29	4.1	28	96.5	1	3.5
Average sleep duration	<6 h	189	25.6	172	91.0	17	9.0
	7–8 h	494	67.0	389	78.8	105	21.2
	≥9 h	54	7.3	37	68.5	17	31.5
Practice of a sport activity	Never	219	29.7	189	86.3	30	13.7
	Rarely	180	24.4	158	87.8	22	12.2
	Once a week	139	18.8	103	74.1	36	25.9
	Several times a week	164	22.2	124	75.6	40	24.4
	Every day	35	4.7	24	68.6	11	31.4
Frequency of alcohol consumption	Never	324	44.0	267	82.4	57	17.6
	Less than once a month	186	25.2	154	82.8	32	17.2
	Between once a month and once a week	196	26.6	156	79.6	40	20.4
	Several times a week	31	4.2	21	67.7	10	32.3
Possible changes in alcohol consumption	No, I never drink alcohol	298	40.4	245	82.2	53	17.8
	No, I kept the same alcohol consumption	188	25.5	141	75.0	47	25.0
	Yes, I have cut down on my drinking a bit	215	29.2	181	84.2	34	15.8
	Yes, I have increased my alcohol consumption a bit	36	4.9	31	86.1	5	13.9
Smoking	No	654	88.8	525	80.3	129	17.7
	Yes	83	11.2	73	87.9	10	12.1
Possible changes in smoking	No, I never smoke	638	86.6	513	80.4	125	19.6
	No, I kept the same smoking level	19	2.6	4	78.9	15	21.1
	Yes, I have cut down on smoking a bit	21	2.8	5	23.8	16	76.2
	Yes, I have increased smoking a bit	59	8.0	54	91.5	5	8.5
Cannabis use	No	698	94.7	563	80.7	135	19.3
	Yes	39	5.3	35	89.7	4	10.3
Possible changes in cannabis use	Never	696	94.4	562	80.7	134	19.3
	No change	25	3.4	22	88.0	3	12.0
	Decrease	8	1.1	7	12.5	1	87.5
	Increase	8	1.1	7	12.5	1	87.5
Date of last consultation with a general practitioner	<12 months	165	22.4	125	75.8	40	24.2
	≥12 months	211	28.6	166	78.7	45	21.3
	Never consulted a general practitioner	362	49.0	307	84.8	55	15.2

**Table 2 ijerph-18-12951-t002:** Educational characteristics.

				Psychological Distress			
		Total		Yes (N = 625, 75.1%)		No (N = 207, 24.9%)	
		N	%	N	%	N	%
Seniority in medical studies	1st year	492	59.4	379	77.0	113	23.0
	2nd and 3rd year	128	15.4	94	73.4	34	26.6
	4th, 5th, and 6th year	209	25.2	152	72.7	57	27.3
Weekly study time (hours)	<10 h	79	10.8	64	19.0	15	81.0
	10–19 h	72	9.9	56	77.8	16	22.2
	20–29 h	63	8.6	47	74.6	16	25.4
	30–39 h	100	13.7	74	74.0	26	26.0
	40–49 h	88	12.1	77	87.5	11	12.5
	>50 h	326	44.8	272	83.4	54	16.6
Sense of mutual support and cooperation	Yes	382	52.5	277	72.5	105	27.5
	No	346	47.5	313	90.5	33	9.5
Impression of recognized work	Yes	286	60.7	192	67.1	94	32.9
	No	442	39.3	398	90.0	44	10.0
Perceived level of stress related to studies	Low to moderate	171	23.5	88	51.5	83	48.5
	High	558	76.5	503	90.1	55	9.9
Face-to-face courses in the last 3 months	No	541	74.3	443	81.9	98	18.1
	Yes, 1 day a week on average	139	19.1	110	79.1	29	20.9
	Yes, 2 days a week on average	32	4.4	27	84.4	5	15.6
	Yes, 3 days per week on average	16	2.2	10	62.5	6	37.5
Hospital internship in the last 3 months	No	439	60.3	369	84.0	70	16.0
	Yes	289	39.7	221	76.5	68	23.5
Internship in COVID-19 care units	No	567	77.9	469	82.7	98	17.3
	Yes	161	22.1	121	75.2	40	24.8
Difficulties in following distance learning courses due to:							
Time management, personal life, or occupational life	No	76	12.7	184	84.4	34	15.6
	Yes	524	87.3	399	88.7	43	11.3
Workload	No	228	38.0	180	78.9	48	21.1
	Yes	372	62.0	343	92.2	29	7.8
Lack of communication with the teaching staff	No	341	56.8	291	85.34	50	14.66
	Yes	259	43.2	232	89.6	27	10.4
Work location	No	378	63.0	326	86.2	52	13.8
	Yes	222	37.0	197	88.7	25	11.3
Lack of communication with other students	No	481	80.2	414	86.0	67	14.0
	Yes	119	19.8	109	91.6	10	8.4
Equipment	No	524	87.3	455	86.8	69	13.2
	Yes	76	12.7	68	89.5	8	10.5
Paid work outside the framework of studies	No	570	78.3	466	81.8	104	18.2
	Yes	158	11.7	124	78.5	34	21.5

**Table 3 ijerph-18-12951-t003:** Factors associated with psychological distress on univariate and multivariate analysis.

		Psychological Distress	
Variables		PR [CI]	Adjusted PR [CI]
Gender	Women (ref: Men)	1.12 [1.02–1.23] *	1.09 [1.00–1.19] *
Age		0.98 [0.96–0.99] *	0.98 [0.95–1.01]
Seniority in the medical studies	1st year medicine main stream (ref: 4th, 5th, and 6th year)	1.11 [1.01–1.21] *	0.92 [0.77–1.12]
Financial difficulties	Yes (ref: NO)	1.11 [1.01–1.22] *	1.03 [0.93–1.15]
Trauma experienced during the COVID-19 crisis	Yes (ref: NO)	1.21 [1.11–1.29] ****	1.10 [1.03–1.16] **
History of anxiety disorder	Yes (ref: NO)	1.12 [1.03–1.22] **	1.11 [1.01–1.20] *
History of depression	Yes (ref: NO)	1.13 [1.03–1.24] *	1.01 [0.91–1.12]
Possible change in alcohol consumption	Yes (ref: NO)	1.08 [1.01–1.16] *	1.12 [1.05–1.20] **
Possible change in smoking	Yes (ref: NO)	1.08 [0.99–1.19]	1.01 [0.92–1.11]
Paid work outside the framework of studies	Yes (ref: NO)	0.90 [0.81–1.00]	0.92 [0.82–1.03]
Sense of mutual support and cooperation	Yes (ref: NO)	0.80 [0.75–0.86] ****	0.92 [0.86–0.98] *
Impression of recognized work	Yes (ref: NO)	0.75 [0.69–0.82] ****	0.84 [0.77–0.91] ***
Hospital internship in the last 3 months	Yes (ref: NO)	0.91 [0.84–0.98] *	/
Internship in COVID-19 care units	Yes (ref: NO)	0.90 [0.81–0.99] *	0.94 [0.82–1.08]
Difficulties in following distance learning courses	Yes (ref: NO)	1.64 [1.39–1.94] ****	1.50 [1.28–1.77] ***
Consultation with a general practitioner	Yes (ref: NO)	0.91 [0.85–0.98] *	0.97 [0.90–1.05]

*p*-value ≤ 0.2; * *p*-value < 0.05; ** *p*-value < 0.01; *** *p*-value < 0.001; **** *p*-value< 0.0001; PR: prevalence ratio; CI: 95% confidence interval.

## Data Availability

The data presented in this study are available on request from the corresponding author. The date are not publicly available due to confidentiality of participants.

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
