# Peer review of "Factors Associated with Psychological Distress in French Medical Students during the COVID-19 Health Crisis: A Cross-Sectional Study"

_ijerph, 2021, doi:10.3390/ijerph182412951_

Round 1

Reviewer 1 Report

This is an interesting study, I have made some recommendations/comments below:

Abstract: I think is not correct to report that the study was conducted on 1814 medical students as reading the manuscript it is clear that 1814 was the eligible population but the number of particpants who completed the questionnaire in full was 699 students...less than the half. I think the Authors should correct the insormation regarding the sample size reported in the abstract.

Introduction: I suggest to the Authors to clearly report the aims of the study and their hypotheses deepening what they wrote (page 2 line 68-69)

Measurements: I suggest to report the psychometric propreprties regarding the GHQ-12 as for example the Chronbach's alpha obtained in the present study.

Results: please correct the editing of page 7 line 145-160

Limits: I suggest to insert a further limit: the very high rate of refusal and drop out that make the present sample a very selected one and have strong implication for the generalizability of the findings.

Author Response

We thank the three reviewers for their suggestions of modifications which contribute to improve the article

REVIEWER 1:

Abstract:

I think is not correct to report that the study was conducted on 1814 medical students as reading the manuscript it is clear that 1814 was the eligible population but the number of particpants who completed the questionnaire in full was 699 students...less than the half. I think the Authors should correct the insormation regarding the sample size reported in the abstract.

Response to reviewer 1:

The summary has been modified to specify the number of students in the eligible population (n=1814), the number of students who completed the questionnaire (n=832) and those who completed it fully (n=699)

Introduction

I suggest to the Authors to clearly report the aims of the study and their hypotheses deepening what they wrote (page 2 line 68-69)

Response to reviewer 1:

The authors have outlined in the introduction the organizational changes in theoretical and practical teaching for medical students brought about by the health crisis. “The spread of the COVID-19 virus has had far-reaching consequences, and the closure of universities has led to the development of online learning, leading to student isolation. Students experiencing higher psychological distress are at a higher risk of academic failure and dropout. In addition, medical students during internship in Covid-19 care unit were involved in the management of patients infected with COVID-19 which may have exposed them to a high emotional burden.”

They outlined the research hypotheses and objectives of this study

We assume that the organizational changes in theoretical (distance learning courses) and practical (internship in COVID 19 care services) teaching caused by the health crisis are associated with psychological distress of medical students

The objectives of the present study were to evaluate the prevalence of psychological distress in medical students during the COVID-19 health crisis and to identify personal, medical and occupational factors associated with psychological distress.

Measurements

I suggest to report the psychometric properties regarding the GHQ-12 as for example the Chronbach's alpha obtained in the present study.

Response to reviewer 1:

The authors added the cronback coefficient of the GHQ-12:”

The cronback coefficient of the GHQ-12 was evaluated by Golberg between 0.82 and 0.86 in general health care (19). The cronback coefficient of the GHQ-12 was evaluated at 0.85 in a population of Malian students(20).  “

  1. Goldberg DP, Gater R, Sartorius N, Ustun TB, Piccinelli M, Gureje O, et al. The validity of two versions of the GHQ in the WHO study of mental illness in general health care. Psychological Medicine. janv 1997;27(1):191‑7.
  2. Yusoff MSB. The validity of two malay versions of the general health questionnaire (ghq) in detecting distressed medical students. ASEAN Journal of Psychiatry [Internet]. 2010 [cité 25 nov 2021];11(2). Disponible sur: http://mymedr.afpm.org.my/publications/44988

Results: please correct the editing of page 7 line 145-160

Response to reviewer 1:

The authors added odds ratios with confidence intervals to the factors associated with psychological distress

“The univariate analysis produced associations between psychological distress with:

–    Female gender PR= 1.12 [1.02-1.23]

–    Age PR= 0.98 [0.96-0.99]

–    1st year of medical school (major) PR= 1.11 [1.01-1.21]

–    Financial difficulties PR=1.11 [1.01-1.22]

–    Psychological trauma during the health crisis COVID 19 PR= 1.21 [1.11-1.29]

–    History of anxiety disorder PR= 1.12 [1.03-1.22]

–    History of depression PR= 1.13 [1.03-1.24]

–    Change in smoking PR= 1.08 [0.99-1.19]

–    Change in alcohol consumption, PR=1.08 [1.01-1.16]

–    Sense of mutual support and cooperation PR=0.80 [0.75-0.86]

–    Impression of recognized work PR=0.75 [0.69-0.82]

–    Hospital internship within the last three months PR= 0.91 [0.84-0.98]

–    Hospital internship on a COVID-19 ward within the last three months PR= 0.90 [0.81-0.99]

–    Experiencing difficulties in online learning. PR= 1.64 [1.39-1.94]

In contrast, psychological distress was not significantly associated with increased cannabis use, or weekly study workload.

Limits: I suggest to insert a further limit: the very high rate of refusal and drop out that make the present sample a very selected one and have strong implication for the generalizability of the findings

Response to reviewer 1:  The authors have added the following sentences as a new limit

Furthermore, 45% of the students enrolled at the Saint-Etienne medical school agreed to participate in this study. Of these, 92% completed the GHQ-12 and 84% completed the questionnaire in full. The refusal and drop-out rates of this study should be considered before generalizing the findings of this study.

Reviewer 2 Report

Thank you for giving me the possibility to review the paper " Factors associated with psychological distress in French medical students during the COVID-19 health crisis: a cross-sectional study". The present study deals with an interesting topic, since it aims to investigate the prevalence of psychological distress and the factors associated with psychological distress among French medical students, during the COVID-19 health crisis. The paper is within the topic of the journal. However, parts of the manuscript need to be improved and I have made suggestions.

Introduction:

  1. Page 2, line 62 to 67

  The authors should state their hypothesis on the basis of previous studies on the prevalence of psychological distress and its associated factors among medical students.

Materials and Methods

  1. Page 2, line 77 to 81

  The authors could better describe and argue the profile of the sample, i.e. the medical students. Are there any peculiarities of the students of the Saint Etienne medical school? What are the specific context factors? I think the answers to these will add more detail to the limitation.

  1. Page 3, line 113 to 116

  The authors excluded only suicidal ideation in main analyses. Have you checked for multicollinearity for other variables such as history of anxiety disorder and that of depression? Why did you put these two histories separately, but not “history of anxiety disorder and/or depression ” into the models?

  1. Page 3, line 117 to 122

  Why do you use multivariate logistic regression? I think you should choose Poisson regression using robust variance estimations because the prevalence of psychological distress was more than 10% in your data. Please read the following articles:

Deddens, J.A.; Petersen, M.R. Approaches for estimating prevalence ratios. Occup. Environ. Med. 2008, 65, 501-506.

Tanji, F.; Kodama, Y. Prevalence of Psychological Distress and Associated Factors in Nursing Students during the COVID-19 Pandemic: A Cross-Sectional Study. Int. J. Environ. Res. Public Health 2021, 18, 10358.

Discussion

  1. Page 10, line 248

  The authors should explain a possibility of selection bias as the response rate in this study is very low. For example, I think you should compared the profile and the percentage of gender of the respondents with those of 1,814 eligible students.

  1. Page 11, line 262 to 264

  Although this study was conducted anonymously, how will you conduct a follow-up survey?

Author Response

We thank the two reviewers for their suggestions of modifications which contribute to improve the article

Reviewer 2

The paper is within the topic of the journal. However, parts of the manuscript need to be improved and I have made suggestions.

Introduction:

  1. Page 2, line 62 to 67

  The authors should state their hypothesis on the basis of previous studies on the prevalence of psychological distress and its associated factors among medical students.

 Response to reviewer 2:

The authors have outlined in the introduction the organizational changes in theoretical and practical teaching for medical students brought about by the health crisis. “The spread of the COVID-19 virus has had far-reaching consequences, and the closure of universities has led to the development of online learning, leading to student isolation. Students experiencing higher psychological distress are at a higher risk of academic failure and dropout. In addition, medical students during internship in Covid-19 care unit were involved in the management of patients infected with COVID-19 which may have exposed them to a high emotional burden.”

They outlined the research hypotheses and objectives of this study

We assume that the organizational changes in theoretical (distance learning courses) and practical (internship in COVID 19 care services) teaching caused by the health crisis are associated with psychological distress of medical students

The objectives of the present study were to evaluate the prevalence of psychological distress in medical students during the COVID-19 health crisis and to identify personal, medical and occupational factors associated with psychological distress.

Materials and Methods

  1. Page 2, line 77 to 81

  The authors could better describe and argue the profile of the sample, i.e. the medical students. Are there any peculiarities of the students of the Saint Etienne medical school? What are the specific context factors? I think the answers to these will add more detail to the limitation.

Response to reviewer 2:

The authors elaborated on the particularity of medical students in the medical school

The target population was students registered at the Faculty of Medicine of Saint Etienne from 1st to 6th year located in the Loire department which recorded the highest incidence rate of COVID-19 in the autumn of 2020 (700 cases per 100,000 inhabitants). This epidemic situation has led to an increase in the demand for care in the hospitals of the Loire. Medical students in hospital internships were involved in the health crisis by participating in the care of patients infected with COVID-19. Students from the 1st to the 3rd year may have been victims of social isolation due to the distance learning course and the implementation of a lockdown period from 30th October 2020 to 15th December 2020.”

The composition of the target population was specified in terms of gender, age and promotion distribution;

 “The average age of the eligible subjects in the study was 21 years [Et=1.17], 66% were women, 65% were 1st year students, 19% were 2nd and 3rd year students and 26% were 4th, 5th and 6th year medical students.

  1. Page 3, line 113 to 116

  The authors excluded only suicidal ideation in main analyses. Have you checked for multicollinearity for other variables such as history of anxiety disorder and that of depression? Why did you put these two histories separately, but not “history of anxiety disorder and/or depression ” into the models?

Response to reviewer 2:

The authors have clarified the search for collinearity of variables in the text by adding the following paragraph:

“A search for multicollinearity was conducted for the following variables: Age ,gender ,financial difficulties, trauma experienced during Covid crisis, history of anxiety disorders, history of depression, changes in alcohol consumption, changes in smoking, paid work outside the framework of studies, sense of mutual support and cooperation, impression of recognized work, hospital internship in the last 3 months, internship in Covid-19 care units, difficulties in following distance learning courses due to, date of last consultation with a general practitioner. Following this analysis, it was decided to remove the internship variable during the last three months due to a significant multicollinearity with other variables. “

It seemed to the authors that there would be a loss of information if “history of anxiety disorders” and  “history of depression “ were combined. For this reason, the authors preferred to keep these two variables in the models.

 Response to reviewer 2:

  1. Page 3, line 117 to 122

  Why do you use multivariate logistic regression? I think you should choose Poisson regression using robust variance estimations because the prevalence of psychological distress was more than 10% in your data. Please read the following articles:

Deddens, J.A.; Petersen, M.R. Approaches for estimating prevalence ratios. Occup. Environ. Med. 2008, 65, 501-506.

Tanji, F.; Kodama, Y. Prevalence of Psychological Distress and Associated Factors in Nursing Students during the COVID-19 Pandemic: A Cross-Sectional Study. Int. J. Environ. Res. Public Health 2021, 18, 10358.

Reponse to reviewer 2:

The authors thank the reviewer for providing references to support the choice of statistical method.  The authors modified the method of statistical analysis with Poisson regression using robust variance estimation to implement the more appropriate analysis in the situation of this study. The values of the prevalence ratios and their confidence intervals are presented in Table 3 and Figure 2.

Discussion

  1. Page 10, line 248

  The authors should explain a possibility of selection bias as the response rate in this study is very low. For example, I think you should compared the profile and the percentage of gender of the respondents with those of 1,814 eligible students.

Response to reviewer 2:

The authors justified the presence of potential selection bias by describing the gender and grade distribution of medical students in the sample and in the source population.

The authors have added the following sentences to the manuscript:

“Besides, the sample in our study has a slightly different distribution from the source population with regard to gender (73% women in the sample, 66% in the population) and years of study (60% first-year students in the sample, 56% in the population). These differences in distribution may contribute to an overestimation of psychological distress. Indeed, according to our study, female gender appears to be a factor positively associated with psychological distress while age appears to be negatively associated with psychological distress. These results are consistent with the study by Maser et al. of Canadian medical students(47).

  1. Page 11, line 262 to 264

  Although this study was conducted anonymously, how will you conduct a follow-up survey?

Response to reviewer 2:

The authors would like to evaluate the impact of national platform of psychological support on the psychological distress of students by means of a new study to assess the prevalence of psychological distress among medical students, one year after the first study. The students received an invitation to participate in this study via their e-mail address. It is therefore possible to send them a new invitation to participate in a new study.

The modalities of participation in the study via their e-mail address were specified in the manuscript.

They received an invitation to participate in this study via their e-mail address.

Round 2

Reviewer 2 Report

Thank you for thoroughly addressing the comments.  Your revisions have helped improve the clarity and readability of the paper.